# Phytochemical Investigation of *Lepionurus sylvestris* Blume and Their Anti-Diabetes Effects via Anti-Alpha Glucosidase and Insulin Secretagogue Activities Plus Molecular Docking

**DOI:** 10.3390/ph16081132

**Published:** 2023-08-10

**Authors:** Sathianpong Phoopha, Surat Sangkaew, Chatchai Wattanapiromsakul, Wanlapa Nuankaew, Tong Ho Kang, Sukanya Dej-adisai

**Affiliations:** 1Department of Pharmacognosy and Pharmaceutical Botany, Faculty of Pharmaceutical Sciences, Prince of Songkla University, Hat Yai 90112, Songkhla, Thailand; sathianpong.p@psu.ac.th (S.P.); surat0647699818@gmail.com (S.S.); chatchai.w@psu.ac.th (C.W.); 2Traditional Thai Medical Research and Innovation Center, Faculty of Traditional Thai Medicine, Prince of Songkla University, Hat Yai 90112, Songkhla, Thailand; 3Graduate School of Biotechnology, Department of Oriental Medicine Biotechnology, College of Life Sciences, Kyung Hee University, 1732, Deogyeong-daero, Giheung-gu, Yongin-si 17104, Gyeonggi-do, Republic of Korea; wanlapa.nuankaew@gmail.com (W.N.); panjae@khu.ac.kr (T.H.K.)

**Keywords:** anti-diabetes, anti-alpha glucosidase, insulin secretagogue, *Lepionurus sylvestris*, phytochemistry, embryonic toxicity

## Abstract

This study presents a phytochemical investigation of *Lepionurus sylvestris* leaf extracts and their anti-diabetic activities. Traditionally, *L. sylvestris* leaves were used as vegetables and food in local recipes, but the root extracts of the plant can also be used in body tonic and erectile dysfunction treatments. Following a preliminary anti-diabetic activity screening test, the 80% ethanolic leaf extract exhibited potent anti-alpha glucosidase activity. So, the leaves’ active components were selected for further investigation. Firstly, the plant was extracted via maceration using lower to higher polarity solvents such as hexane, ethyl acetate, ethanol, and water, respectively, to obtain the four crude extracts. Then, the phytochemicals contained in this plant were investigated via classical column chromatography and spectroscopy techniques. Anti-diabetic activity was evaluated via anti-alpha glucosidase and insulin secretagogue assays. The results showed that five compounds were isolated from the fractionated ethanolic leaf extract: interruptin A; interruptin C; ergosterol; diglycerol; and 15-16-epoxy-neo-cleoda-3,7(20),13(16),14-tetraene-12,17:18,19-diolide, a new diterpene derivative which is herein referred to as lepionurodiolide. Interruptin A and the new diterpene derivative exhibited the greatest effect on anti-alpha glucosidase activity, showing IC_50_ values of 293.05 and 203.71 μg/mL, respectively. Then, molecular docking was used to study the sites of action of these compounds. The results showed that interruptin A and the new compound interacted through H-bonds with the GLN279 residue, with a binding energy of −9.8 kcal/mol, whereas interruptin A and C interacted with HIS280 and ARG315 a with binding energy of −10.2 kcal/mol. Moreover, the extracts were investigated for their toxicity toward human cancer cells, and a zebrafish embryonic toxicity model was used to determine herbal drug safety. The results indicated that ethyl acetate and hexane extracts showed cytotoxicity to both Hela cells and human breast adenocarcinomas (MCF-7), which was related to the results derived from using the zebrafish embryonic toxicity model. The hexane and ethyl acetate presented LC_50_ values of 33.25 and 36.55 μg/mL, respectively, whereas the ethanol and water extracts did not show embryonic toxicity. This study is the first of its kind to report on the chemical constituents and anti-diabetic activity of *L. sylvestris*, the leaf extract of which has been traditionally used in southern Thailand as a herbal medicine and food ingredient.

## 1. Introduction

Diabetes mellitus (DM) is a chronic metabolic syndrome which leads to the malfunction of glucose homeostasis in blood circulation. Abnormalities in the pancreas significantly reduce insulin, which plays a key role in glucose metabolism. The effects of high-level glucose concentrations in the bloodstream might cause tissue damage, especially with respect to blood vessels, the nervous system, and wound healing processes. In 2021, the International Diabetes Federation reported that 573 million adults around the world were living with diabetes, and estimates state that the number of patients will increase to 783 million by 2045. The Division of Non-Communicable Diseases of Thailand stated that the number of diabetic patients in the nation was 0.94 million in 2017; however, figures seem to be continuously rising. Traditionally, undeveloped and developing countries have used herbal medicines as alternatives for combatting diabetes for a long time, and various types of remedies have been prepared based on traditional wisdom. Herbal medicines are meant to improve one’s health and prevent and reduce the severity of diseases. Prior to human applications, the efficacy of using of herbal medicines and their mechanisms of action and toxicity need to be clarified.

The chemicals of natural products are considered notable resources for researching the biological activities and the benefits and functions of substances that can be used in herbal medicines. *L. sylvestris* (Opiliaceae) or “Mak-Mok” (common Thai name) is a plant found in southern Thailand. In traditional herbal remedies, the root of this plant is used as a body tonic and for erectile dysfunction treatment. The leaf and stem of this plant can be used in food and herbal remedies as diuretics and aphrodisiacs, and for the treatment of kidney stone diseases. Additionally, the aerial part of the plant can be used as an ingredient in rejuvenating recipes and has demonstrated anti-microbial and anti-oxidant activity in prior studies [1,2]. Indeed, some articles have reported the biological activities of *L. sylvestris* leaf extracts; for example, DPPH radical scavenging assays have shown its anti-oxidant activity, and it has also been shown to exert anti-bacterial activity on *H. pyroli* strains. The root extracts of this plant also reportedly exert anti-oxidant activity and cytotoxic effects on brine shrimp [3,4,5]. Despite this, the plant has not been recommended for use in traditional Thai anti-diabetes medicines, and whether the plant exerts anti-diabetic activity has not been reported. Due to the fact that a preliminary study on anti-alpha glucosidase activity showed the potent effect of this plant extract, this plant was selected for further phytochemical study. Subsequently, after studying the plant’s extracts and isolated compounds, it was shown that *L. sylvestris* has the potential to exert strong anti-alpha glucosidase and insulin secretion activities. Hence, this study on the plant extracts and isolated compounds of *L. sylvestris* will be the first scientific report to elucidate the plant’s anti-diabetic activity and mechanisms of action. The results of this research study will hopefully lead to the use of this plant in the development of food products and herbal medicines intended for diabetes prevention.

## 2. Results

### 2.1. Phytochemical Investigation from Ethanolic Leaf Extract of L. sylvestris

#### 2.1.1. Interruptin A (**1**)

The compound (2.40 mg) was obtained as a yellow-white powder dissolved in chloroform. The UV absorption in chloroform exhibited a λmax of 290 nm, as shown in the Appendix A. The IR spectra of this compound presented peaks at 3361 cm^−1^ (-OH), 1705 cm^−1^ (C=O), 1435 cm^−1^ (-CH_2_), and 1140 cm^−1^ (C-O), as shown in the Appendix A. The molecular ion peak was found at 399.1321 (M-H), as shown in the Appendix A. The ^1^H NMR (500 MHz in CHCl_3_-*d*) spectrum exhibited δ_H_ (ppm) values of 12.90 (1H, *s*), 7.57–7.62 (2H, *m*), 7.42–7.48 (1H, *m*), 7.15–7.30 (3H, *m*), 5.29 (1H, *s*), 3.25 (1H, *t*, J = 7.0 Hz), 2.93 (1H, *t*, J = 7.5 Hz), and 2.25 (3H, *s*), as shown in the Appendix A. The ^13^C NMR (125 MHz in CHCl_3_-*d*) spectrum showed the δ_C_ (ppm) at 206.1, 164.6, 159.4, 157.0, 156.8, 153.2, 141.1, 135.9, 130.6, 130.0, 128.4 (double ratio), 127.5, 126.0, 112.7, 106.9, 99.8, 46.8, 30.2, and 7.5 as shown in the Appendix A. The correlation of COSY and HMBC was used to confirm the substitution of the structure, as shown in the Appendix A. The chemical shift data were compared to a previous report on interruptin A, as shown in Figure 1 [6].

#### 2.1.2. Interruptin C (**2**)

The compound (1.30 mg) was obtained as a yellow, amorphous, and clearly dissolved in chloroform and ethyl acetate. The UV absorption in chloroform exhibited a λmax at 291 nm, as shown in the Appendix A. The IR spectra showed peaks at 3502 cm^−1^ (-OH), 2298 cm^−1^ (C=C), 1698 cm^−1^ (C-H), 1473 and 1456 cm^−1^ (-CH_2_), and 1070 cm^−1^ (C-O), as shown in the Appendix A. The molecular ion peak was found at 399.1291 (M-H), as shown in the Appendix A. The ^1^H NMR (500 MHz in CHCl_3_-*d*) spectrum exhibited the δ_H_ (ppm) at 12.97 (1H, *s*)*,* 7.38–7.46 (9H, *m*), 7.30–7.34 (2H, *m*), 6.04 (1H, *s*), 3.14 (1H, *dd*, J = 17.5, 4.5 Hz), 2.96 (1H, *dd*, J = 13, 3 Hz, and 2.25 (3H, *s*), as shown in the Appendix A. The ^13^C NMR (125 MHz in CHCl_3_-*d*) spectrum showed the δ_C_ (ppm) at 197.2, 160.8, 160.2, 159.7, 155.7, 138.7, 137.7, 129.0 (double ratio), 128.9 (double ratio), 128.4 (double ratio), 127.7, 113.1, 105.3, 78.8, 43.2, and 7.86, as shown in the Appendix A. The correlation of COSY and HMBC was used to confirm the substitution of the structure, as shown in the Appendix A. The chemical shifts were compared to a previous report on interruptin C, as shown in Figure 1 [6].

#### 2.1.3. Ergosterol (**3**)

The compound (1.25 mg) was obtained as a white crystal dissolved in chloroform and ethyl acetate. The UV absorption in chloroform exhibited a λmax at 254 nm (Appendix A) and presented a blue fluorescent spot under UV at 366 nm in thin-layer chromatography. The IR spectra showed peaks at 3503 cm^−1^ (-OH), 2932 cm^−1^ (C-H), and 1699 cm^−1^ (C=O), as shown in the Appendix A. The ^1^H NMR (500 MHz in CHCl_3_-*d*) spectrum exhibited the characteristics of the steroid structure by showing the δ_H_ (ppm) at 0.61 (3H, *s*, H-18), 0.84 (3H, *d*, J = 7.0 Hz, H-26), 0.85 (3H, *d*, J = 7.0 Hz, H-27), 0.92 (3H, *d*, J = 7.0 Hz, H-28), 0.94 (3H, *s*, H-19), and 1.01 (3H, *d*, J = 6.6 Hz, H-21); an oxygenated proton signal at δ 3.62 (1H, m, H-3); and olefinic proton signals at 5.15 (2H, *m*, H-22), 5.36 (1H, *dd*, J = 16.0, 7.5 Hz, H-23), and 5.55 (1H, *dd*, J = 5.7, 2.5 Hz, H-6), as shown in the Appendix A. The chemical shifts were compared to a previous report on ergosterol, as shown in Figure 1 [7].

#### 2.1.4. Diglycerol (**4**)

The compound (1.50 mg) was obtained as a white needle dissolved in DMSO. The IR spectra presented peaks at 3399 cm^−1^ (-OH), 2970 cm^−1^ (-CH), 2902 cm^−1^ (-CH_2_), and 1080 cm^−1^ (C-O), as shown in the Appendix A. The ^1^H NMR (500 MHz in DMSO-*d_6_*) spectrum exhibited the δ_H_ (ppm) at 4.37 (1H, *d*, J = 5.5 Hz), 4.28 (1H, *t*, J = 5.5 Hz), 4.11 (1H, *d*, J = 7 Hz), 3.50 (1H, *t*, J = 7.5 Hz), and 3.35 (1H, *dt*, J = 5.5, 11 Hz), as shown in the Appendix A. The ^13^C NMR (125 MHz in DMSO-*d_6_*) spectrum showed the δ_C_ (ppm) at 71.49 (double ratio), 69.88, and 64.00 (double ratio), as shown in the Appendix A. The chemical shifts were compared to a previous report on diglycerol, as shown in Figure 1 [8].

#### 2.1.5. The New Diterpene Derivative: 15-16-epoxy-neo-cleoda-3,7(20),13(16),14-tetraene-12,17:18,19-diolide (Lepionurodiolide) (**5**)

The new compound (5.15 mg) was obtained as a light-yellow power dissolved in chloroform. The UV absorption in chloroform exhibited a λmax at 286 nm, as shown in the Appendix A. The IR spectra presented peaks at 3502 cm^−1^ (-OH), 2953 (C-H), 2298 cm^−1^ (C=C), 1758 cm^−1^ (C=O), 1508 and 1436 cm^−1^ (-CH_2_), and 1153 cm^−1^ (C-O), as shown in the Appendix A.

The ^1^H NMR (500 MHz in CHCl_3_-*d*) spectrum showed the characteristic of the furan proton by exhibiting the δ_H_ (ppm) at δ 6.33 (1H, *dd*, J = 1.9, 1 Hz, H-14), 7.38 (1H, broad *d*, J = 1 Hz, H-15), and 7.37 (1H, *t*, J = 1 Hz, H-16), and showed the methylene proton at 4.25 (1H, *d*, J = 9.1 Hz, H-19) and 4.21 (1H, *dd*, J = 9.1, 1 Hz, H-19). The chemical shifts of the methine proton linked the furan ring and the 6-membered lactone ring at δ 5.51 (1H, *dd*, J = 1.5, 1 Hz, H-12). The other signals showed the characteristic of the alicyclic proton at 1.78 (1H, *m*, H-1), 1.36 (1H, *m*, H-1), 2.43 (1H, *m*, H-2), 2.18 (1H, *m*, H-8), 2.08 (1H, *m*, H-10), 2.19 (1H, *m*, H-2), 2.36 (1H, *m*, H-6), 2.25 (1H, *m*, H-6), 2.55 (1H, *m*, H-9), 2.36 (1H, *m*), 2.75 (1H, *dd*, J = 13.7, 8.6 Hz, H-11), 2.56 (1H, *m*, H-11), and 6.67 (1H, *dd*, J = 2.3, 1 Hz, H-3), and those of the methylene proton in the 6-membered lactone ring at δ 5.01 (1H, *d*, J = 1.7, 1 Hz, H-20) and 4.78 (1H, *d*, J = 1.1, 1 Hz, H-20), as shown in the Appendix A. The correlation between protons (Figure 2) in the structure was observed via ^1^H-^1^H COSY spectrometry, as shown in the Appendix A.

The ^13^C NMR (125 MHz in CHCl_3_-*d*) spectrum revealed 20 carbon signals. It exhibited the δ_C_ (ppm) assignable to six methylene at 21.4 (C-1), 27.7 (C-2), 26.4 (C-6), 53.4 (C-11), 73.4 (C-19), and 113.2 (C-20); eight methines at δ 133.7 (C-3), 43.6 (C-8), 44.0 (C-9), 32.5 (C-10), 71.5 (C-12), 108.4 (C-14), 140.0 (C-15), and 144.0 (C-16); four quaternary carbons at δ 137.0 (C-4), 48.8 (C-5), 147.3 (C-7), and 124.7 (C-13); and two lactone carbonyls at δ 176.7 (C-17) and 169.4 (C-18). The analysis of these ^13^C NMR data indicated the characteristic signals for a neo-cleodane diterpenoids of two quaternary carbons at δ 48.8 (C-5) and 44.0 (C-9); a methine at 32.5 (C-10); an oxygenated methylene at δ 73.4 (C-19); and a substituted furan ring at 124.7 (C-13), 108.4 (C-14), 140.0 (C-15), and 144.0 (C-16), as shown in the Appendix A. The correlation between protons and carbons (Figure 2) in the structure was observed via ^1^H-^13^C HMBC spectrometry, as shown in the Appendix A.

ESI-negative mass spectroscopy was used to confirm the structure; the molecular ion peak at *m*/*z* 339.1238 [M-H] (calcd. 339.1240) established the molecular formula C_20_H_20_O_5_, as shown in the Appendix A. The interpretation of the compound was significantly clarified as a new compound of diterpenoids derivative via the detailed analysis of ^1^H-^1^H COSY, HMBC, and NOESY as 15-16-epoxy-neo-cleoda-3,7(20),13(16),14-tetraene-12,17:18,19-diolide. The stereochemistry of the new compound was proven via NOESY spectrometry, as shown in the Appendix A. The C-10 that resonated at 32.5 ppm indicated that H-10 is β-H. The NOESY spectrum H-10 resonated at 2.08 ppm, showing a correlation with H-9, which resonated at 2.55 ppm. H-9 showed a correlation, resonating at 2.18 ppm (H-8) and 5.51 ppm (H-12). All data suggested that H-10, H-9, H-8, and H-12 are β-H, as shown in Figure 1 and Figure 2. The compound was named using the IUPAC rule as 15-16-epoxy-neo-cleoda-3,7(20),13(16),14-tetraene-12,17:18,19-diolide, also called “lepionurodiolide”.

### 2.2. Total Phenolic and Total Flavonoid Contents

The phenolics and flavonoids contents of *L. sylvestris* leaf extracts with hexane, ethyl acetate, ethanol, and water were determined, as shown in Figure 3. The ethanolic extract exhibited the highest total phenolic and total flavonoid compounds, followed by ethyl acetate, hexane, and water extracts, respectively.

### 2.3. Anti-Alpha Glucosidase and Mechanism of Action

The plant extracts were screened on anti-alpha glucosidase at the concentration of 2 mg/mL to discover the most effective extract, which will be used for further investigating its phytochemistry and mechanism of action. The result presented that the ethanolic leaf extract was the only sample exhibiting a potent alpha glucosidase inhibition of 95.77 ± 0.63%, while the frontrunner was water extract, with 57.99 ± 1.42%, which was compared to the positive control, acarbose, with 88.18 ± 0.53% (Table 1).

The mechanism of action of those two effective extracts was evaluated via Michaelis–Menten and Lineweaver–Burk kinetic plots, which are graphically exhibited in Figure 4. The result indicated that the ethanolic extract inhibited the enzyme activity through an uncompetitive reaction, whereas the water extract inhibited it via a competitive mechanism. The Ki of water extract was calculated as 0.91 mM and compared to that of acarbose, which was 0.23 mM.

### 2.4. Computer Molecular Docking of the Effective Compounds

Molecular docking of the new diterpene derivative (Figure 5D) was performed to investigate its mechanisms of action. The results presented that the H-bond of the new compound interacted with GLN279 residue with a binding energy of −9.8 kcal/mol. Interruptin A and C bound to HIS280 and ARG315 at the carbonyl group position with the same binding energy, −10.2 kcal/mol. In addition, interruptin B, the structure related to interruptin A and C, was graphically computed to compare its binding energy with the isolated compounds. The result indicated that the three structures of interruptin compounds significantly bound to the same site of the enzyme at HIS 280, where they interacted with the catalytic pocket of alpha glucosidase, playing an important role in the hydrolysis of the glycosidic bond, as shown in Figure 5. 

### 2.5. Insulin Secretagogue Activity

The isolated compounds of *L. sylvestris* were screened on insulin-secreting rat insulinoma (INS-1E) cells to evaluate the insulin secretion at a concentration of 100 μg/mL. The results indicated that these compounds significantly stimulated insulin secretion activity from the beta cells compared to the control. Interruptin A, interruptin C, ergosterol, and diglycerol potentially promoted the insulin secretion, and we calculated the total insulin secretion at 102.52, 102.05, 102.45, and 102.21 μg/L, respectively, whereas the new diterpene compound slightly induced the secretory activity at 52.06 μg/L. Glibenclamide was used as the positive standard in this experiment, which induced the activity at 104.57 μg/L, and we compared it to the negative control (2.5 mM glucose in RPMI-1640 medium), as shown in Figure 6.

### 2.6. Cytotoxic Effect on Human Cancer Cells and Toxicity in Zebrafish Model

In this study, cytotoxic activity was determined, at a concentration of 25 µg/mL, on three human cancer cell lines, A549 (human lung adenocarcinoma), MCF-7 (human breast adenocarcinoma), and Hela (human cervix adenocarcinoma), and one normal human cell, HGF (human gingival fibroblast). The most effective extract on cytotoxic activity was ethyl acetate extract, which potentially inhibited the proliferation of MCF-7 and HeLa. In contrast, all extracts of *L. sylvestris* presented slightly cytotoxic activity on the A549 cell line. So, from these results, it seemed that *L. sylvestris* extracts affected the cancer cell lines. However, due to ISO 10993-5, the non-cytotoxic agents should provide a cell viability above 80%. So, those compounds that presented activity of 80–60%, 60–40% and below 40% of the cell viability on the normal cell line (HGF) were generally considered weak, moderate, and strong cytotoxic substances, respectively [9]. Hence, the hexane and ethyl acetate extracts were formally categorized as moderate cytotoxic extracts, while the ethanol and water extracts exhibited weak cytotoxic activity on the HGF cells. The results are shown in Table 2.

In addition, the toxicity of *L. sylvestris* leaf extracts on an animal model was evaluated using a zebrafish embryonic model, to which we administered the median lethal concentration (LC_50_). In brief, the zebrafish embryos were treated with hexane, ethyl acetate, ethanol, and water extracts with different concentrations, and DMSO was used as a negative control. The LC_50_ values of hexane, ethyl acetate, and ethanol extracts were graphically calculated as 33.26, 36.55, and 345.9 µg/mL, respectively. The water extract showed an LC_50_ greater than 1000 µg/mL, which was estimated to be a non-toxic extract. Thus, the hexane and ethyl acetate extracts potentially exhibited embryonic toxicity on the zebrafish model, which is related to cytotoxic effects on human cancer cells, as shown in Table 3 and Figure 7.

The physical appearances of zebrafish embryonic model are graphically exhibited in Figure 8. The hatching rate 48 h after the fertilization of the embryos was calculated using the larva viability, which was tested using different concentrations of *L. sylvestris* leaf extracts (10–1000 µg/mL). The result indicated that the increase in the concentration significantly declined the hatching rate. The hexane and ethyl acetate extracts inhibited the hatching rate by less than 50% at the concentration of 50–1000 µg/mL, compared to control group, which inhibited the rate by approximately 80, while ethanol extract potentially decreased the hatching rates at a concentration of 400 µg/mL. Additionally, the water extract significantly presented the highest hatching rate and also decreased the rate at 800 µg/mL. The malfunction and abnormality of larvae were monitored by measuring their length from head to tail and their heart rate. The result showed that all extracts of *L. sylvestris* showed a slight difference in the result in each concentration when compared to the normal control and negative control, and we also did not find any abnormalities in the high-concentration groups, as shown in Figure 8.

## 3. Discussion

The extracts of *L*. *sylvestris* leaves were investigated for anti-diabetic activity, and several experiments were carried out to support their biological activities. The ethanol leaf extract and its anti-diabetic activities were the focus of the phytochemical investigation due to its preliminary result on the anti-alpha glucosidase enzyme. The result indicated that ethanol and water extracts significantly inhibited the activity of the alpha glucosidase enzyme via uncompetitive and competitive mechanisms, respectively. The Ki of the mechanism being moderately higher than that of acarbose that could be explained by the fact that these extracts interacted with low affinity to the enzyme. The activity of the ethanol extract presented a very low IC_50_ value, which could be caused by the phenolic compounds in the extract.

The isolated compounds exhibited as enzyme inhibitors were categorized as flavonoid and diterpenes groups. The flavonoids, which were characterized as quercetin, showed the best anti-alpha glucosidase activity, compared to the other flavonoid structures [10]. The substitutions played a key role to reduce or enhance the activity of the inhibitors; in particular, hydroxylation in the structure played a powerful role in enhancing the performance of the inhibitors. Methylation or methoxylation affected the activity depending on the situation, but most of the methylation and methoxylation slightly declined the activity. The glycosylation of the flavonoids reduced the performance of the inhibitor via the substitution sites and the class of sugar molecules. The reduction in anti-alpha glucosidase activity after glycosylation was caused by the increase in molecular mass and polarity, and the transfer to the nonplanar compound. After the hydroxyl group was replaced by a sugar molecule, steric hindrance took place, which created a weak binding interaction between the flavonoids and the alpha glucosidase enzyme [11]. The in vitro study on anti-alpha glucosidse activity of the effective compounds was confirmed through molecular docking experiments. The result predicted that these effective compounds (interruptin A, B, and C) interacted with HIS280 and ARG315, which was specified as catalytic pocket of alpha glucosidase with binding energy at −10.2 kcal/mol. This site played a key role in the hydrolysis of the glycosidic bond of the polysaccharide chain, whereas the new diterpene derivative created a H-bond in the GLN279 and ARG315 with binding energy at −9.8 kcal/mol. The binding energy was related to the IC_50_ of these compounds; the new diterpene was the most effective compound, with a low binding energy. The stimulation of insulin secretion activity is a combination mechanism of anti-diabetic treatment that has already been tested in this study. The isolated compounds significantly induced insulin secretion from beta cells more than the control and the positive standard. The mechanism of these compounds had not been investigated. 

The cytotoxic effect on human cancer cells and embryonic toxicity on the zebrafish model were used to predict the benefit and drug safety of *L. sylvestris.* The results suggested that the hexane and ethyl acetate extracts should be used to further study on anti-cancer agents, due to their high toxic effects on some cancer cell lines, whereas the ethanol and water extracts exerted anti-diabetic activity, with non-toxic effects and malfunction in the embryonic toxicity model.

Unfortunately, these isolated compounds appeared in a trace amount; they were not abundant enough for further study in other anti-diabetic models. However, most traditional uses of this plant are in the form of food ingredients or water extraction. Hence, further studies using this plant extracts are still interesting. So, the future research plan of this plant will be focused on diabetic zebrafish models using crude extracts and some isolated compounds available from commercial trading.

## 4. Materials and Methods

### 4.1. Plant Material and Extraction Method

*L. sylvestris* was collected from Rattaphum district, Songkhla province, Thailand. The plants were identified by Mrs. Pranee Rattanasuwan, the scientist, and deposited at the Department of Pharmacognosy and Pharmaceutical Botany, Faculty of Pharmaceutical Sciences, Prince of Songkla University, Thailand, specimen number SKP No. 131 12 19.

Fresh *L. sylvestris* leaves were cleaned with water and dried at 50 °C for 72 h. Dried leaves were coarsely ground and macerated 3 times with various polarity solvents. Initially, powdered leaves (4616 g) were macerated by soaking in hexane solution for 3 days; then, the solution was filtered and evaporated with a rotary evaporator at 55 °C to obtain the hexane extract. Afterward, the marc was macerated with hexane another 2 times to increase the amount of extract. Next, the marc, after hexane extraction, was further extracted with ethyl acetate and ethanol, respectively, following the same procedure of hexane extraction. Finally, the dried residue of the leaves was boiled in hot water at 80 °C for 8 h, and the solution was evaporated to obtain the water extract. The four solvent extracts were obtained from lower to higher solvent polarity as shown in Figure 1. The crude extracts of *L. sylvestris* obtained from these solvents were calculated for extraction yield, followed by hexane extract (71.03 g, 1.53 % *w*/*w*), ethyl acetate extract (41.23 g, 0.89% *w*/*w*), ethanol extract (281.03 g, 6.08% *w*/*w*), and aqueous extract (279.76 g, 6.05% *w*/*w*), respectively.

### 4.2. Phytochemical Investigation and Structure Elucidation Techniques

The phytochemical compositions of the plant extracts were separated using classical column chromatographic techniques. Then, the isolated compounds were elucidated via ^1^H and ^13^C Nuclear Magnetic Resonance spectrometry (1D and 2D NMR) monitoring with a Fourier Transform NMR Spectrometer (^1^H-NMR 500 MHz and ^13^C-NMR 125 MHz), model UNITY INNOVA, Varian (Scientific Equipment Center, Prince of Songkla University). The UV spectra were obtained using a Spectronic Genesys 6 UV-Visible Spectrometer, Thermo Scientific, Thermo Electron Corporation, and IR (KBr disc) spectra were obtained using a Perkin Elmer FT-IR Spectrum One Spectrometer (Department of Pharmaceutical Chemistry, Faculty of Pharmaceutical Sciences, Prince of Songkla University).

### 4.3. Total Phenolics and Total Flavonoids Content

Total phenolics and total flavonoids assays were obtained via Folin–Ciocalteu’s and aluminum chloride colorimetric methods, which were used to evaluate the chemical composition in the crude extracts [12,13]. Technically, the total phenolic content was obtained by mixing of 100 μL of samples, 500 μL of 10% *v/v* Folin–Ciocalteu’s reagent, and 400 μL of 1 mM sodium bicarbonate. Then, the reaction was incubated at room temperature for 30 min, and we measured the color with the UV spectrometer at 765 nm. Gallic acid was used to create the standard calibration curve for total phenolic content evaluation. The determination of the total flavonoid content was tested using the aluminum chloride colorimetric method, which was composed of 100 μL of 10% *w/v* of aluminum chloride, 100 μL of 1 M potassium acetate, 1500 μL ethanol, and 500 μL of samples. Then, the reaction was incubated for 30 min, and we measured the color with the UV spectrometer at 415 nm. Quercetin was used to create the standard calibration curve for total flavonoid content evaluation.

### 4.4. Anti-Alpha Glucosidase Activity and Mechanism of Action

The anti-alpha glucosidase activity was determined by following Phoopha et al.’s method [14]. The assay evaluated the yellow product of *N*-para-nitrophenol, which was hydrolyzed by the alpha glucosidase enzyme at 405 nm. The reaction was prepared using a phosphate buffer (pH 7.0), sample (was dissolved in DMSO), *N*-*para*-nitrophenol as substrate, and alpha glucosidase enzyme solution. The test was evaluated by measuring the velocity of the reaction per minute. Afterward, the effective samples were evaluated for the IC_50_ and mechanism of action to figure out the inhibitory function using Michaelis–Menten and Lineweaver–Burk kinetic plots [15].

### 4.5. Molecular Docking

The structure of alpha glucosidase (PDB ID: 3A4A) was obtained from the RCSB Protein Data Bank. The X, Y, and Z centers of the grid maps were 21.275, −0.741, and 18.635, respectively. The molecular binding interaction of the isolated compounds with alpha glucosidase was performed using AutoDock Vina (version 1.1.2). The 3D structure of the new isolated compound was created via ChemDraw Ultra 12.0 and Open Babel GUI. Discovery Studio 2019 was used for removing ligands and water molecules. Hydrogen atoms and charges were then added using the AutoDock tool. The grid maps were generated with a default spacing of 0.375 Å and a 50 × 50 × 50 grid box size. The molecular bonds of the isolated compound were set to rotatable. All torsions were also allowed to rotate. The interactions of molecular docking were analyzed using Discovery Studio 2019. The lowest binding energy was selected as the best affinity of the molecular interaction [16,17].

### 4.6. Insulin Secretagogue Activity

The experiment was slightly modified from previous reports. In brief, the insulin secretion was determined by testing insulin-secreting rat insulinoma (INS-1E) cells, supported by Prof. Dr. Michael Wink, Institute of Pharmacy and Molecular Biotechnology (IPMB), University of Heidelberg, Germany. The cells were cultured in RPMI-1640 liquid medium and supplementary nutrients for cell culture under the condition of a 37 °C humidified incubator containing 5% CO_2_. The cells were seeded into 24-well plates at 2 × 10^5^ cells. Later, the cells were treated with glucose-free culture media to stop energy consumption in the cell metabolism. Afterward, the minimum concentration of glucose in RPMI-1640 was added to the samples for 24 h. Finally, the insulin secretion was measured for insulin releasing by ELISA technique [18].

### 4.7. Cytotoxic Effects on Human Cancer Cells and Toxicity in Zebrafish Model

The toxicity in human cancer cells was tested on a human breast carcinoma cell line (MCF-7), human cervix adenocarcinoma cell line (HeLa), and human lung carcinoma cell line (A549), which were compared to a human gingival fibroblast cell line (HGF) as the normal cells. These cells were cultured in DMEM liquid medium, supporting supplementary nutrients for cell culture. The cells were raised in the same conditions as INS-1E. A sulforhodamine B assay was used to evaluate the cytotoxic effects of the samples, following a previous report [19]. In brief, the cancer cells were seeded into 96-well plates at 5 × 10^3^ cells and incubated for 24 h. Later, we added the test samples at the final concentration of 25 μg/mL and left the experiment in the incubator for 72 h. After that, the cancer cells were fixed with cold 10% trichloro acetic acid for an hour and then we washed the acid with water. The dried plate was dyed with 0.4% SRB in 1% acetic acid for 30 min, and we washed the color with 1% acetic acid 4 times. The SRB color was dissolved in the cancer cells by adding 10 mM tris-base and measuring with a microplate reader at 492 nm. 

The in vivo toxicity was examined in a zebrafish model following Ko et al.’s method [20]. This experiment was performed at the Graduate School of Biotechnology, Department of Oriental Medicine Biotechnology, College of Life Sciences, Kyung Hee University, Republic of Korea (ethical certificate number KHGASP-21-230), following OECD TG 236 for the Fish Embryo Acute Toxicity (FET) test [21]. In brief, the fertilized embryos of zebrafish were collected and raised for testing with samples at a concentration between 10 and 1000 µg/mL, which was dissolved in DMSO. The result of healthy and dead fish was observed under a microscope. LC_50_ values were calculated via non-linear regression using statistical software.

## 5. Conclusions

The anti-diabetic activity of *L. sylvestris* leaf extracts was evaluated in in vitro, in vivo, and in silico experiments, which were used to clarify the anti-diabetic activity and safety of *L. sylvestris.* The results showed that the fractionated ethanolic leaf extracts of *L. sylvestris* strongly exhibited an effective anti-alpha glucosidase activity through an uncompetitive mechanism. Interruptin C and the new diterpene compounds which were investigated from fractionated ethanolic extracts also inhibited enzyme activity. Moreover, their isolated compounds, interruptin A, interruptin C, ergosterol, and diglycerol, significantly stimulated insulin secretion from beta cells of the INS-1E cell line. The information on the herbal drug safety was predicted from the abnormality and malfunction of a zebrafish embryonic model, which did not present any malfunction or deformities in the larvae.

## Data Availability

Data sharing is not applicable.

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
