# Peer review of "Phytochemical Investigation of Lepionurus sylvestris Blume and Their Anti-Diabetes Effects via Anti-Alpha Glucosidase and Insulin Secretagogue Activities Plus Molecular Docking"

_pharmaceuticals, 2023, doi:10.3390/ph16081132_

Round 1
Reviewer 1 Report
Dear authors.
Please find below comments and observations on your manuscript. You should make changes and update the document.
1) The introduction should include ethnobotanical, entnomedicinal or ethnopharmacological references to the traditional use and medicinal effects of the plant.
2) The authors mention that there are few studies on the plant in question. When searching, some references were found that are not in the manuscript. Therefore, reference should be made to what is currently available on the plant.
Lepionurus sylvestris seedling container cultivation method. By: Chen, Jinyan; Zhao, Zhiheng; Li, Kaixiang; Liang, Wenhui; Liao, Jianming; Zeng, Xiangyan. China, CN105918048 A 2016-09-07 | Language: Chinese, Database: CAplus
Seedling cultivation substrate for Lepionurus sylvestris and preparation method thereof. By: Zhao, Zhiheng; Li, Kaixiang; Liang, Wenhui; Zhu, Changsan; Liao, Jianming; Li, Baocai. China, CN108184604 A 2018-06-22 | Language: Chinese, Database: CAplus
Antimicrobial activities of some Thai traditional medical longevity formulations from plants and antibacterial compounds from Ficus foveolata. By: Meerungrueang, W.; Panichayupakaranant, P. Pharmaceutical Biology (London, United Kingdom) (2014), 52(9), 1104-1109 | Language: English, Database: CAplus and MEDLINE
Antioxidant and lifespan-extending effects of a rejuvenating thai traditional polyherbal remedy (Phy-Blica-O) in Caenorhabditis elegans. By: Chanthasri, Wipawee; Aan, Goon Jo; Singkonpong, Ngamsiri; Sudkhaw, Teerawat; Maneenoon, Katesarin; Limsuwan, Surasak; Sanpinit, Sineenart; Wetchakul, Palika; Chusri, Sasitorn. Tropical Journal of Natural Product Research (2021), 5(9), 1554-1568 | Language: English, Database: CAplus
The first mistletoes: Origins of aerial parasitism in Santalales. By: Vidal-Russell, Romina; Nickrent, Daniel L. Molecular Phylogenetics and Evolution (2008), 47(2), 523-537 | Language: English, Database: CAplus and MEDLINE
Study on the total flavonoids content determination of the Lepionurus sylvestris BL. and its antioxidant activity. By: Zhu, Cheng-hao; Tang, Jian-min; Wei, Xiao; Gao, Li-mei; Zou, Rong; Qin, Hui-zhen; Zhu, Hong-jie, Shipin Yanjiu Yu Kaifa (2019), 40(20), 168-175 | Language: Chinese, Database: CAplus
Lepionurus sylvestris bl. seedling raising method for promoting root system development through root cutting treatment. By: Zhu, Changsan; Chen, Jinyan; Zhao, Zhiheng; Li, Kaixiang; Liang, Wenhui; Huang, Kaishun; An, Jiacheng. China, CN110073886 A 2019-08-02 | Language: Chinese, Database: CAplus
3) All sections of the methodology should be written in detail. The methodology must be rewritten.
4) The discussion and analysis of the results can be more extensive. It should be reviewed and updated.
5) At the end of the discussion section you should include a paragraph or paragraphs with the strengths and weaknesses of your work.
6) What is the name of the new diterpene? The IUPAC name of the new compound must be included.
Authors make changes and update the manuscript.
Regards
Reviewer
Author Response
Reply to Reviewer 1
Comments and Suggestions for Authors
Dear authors.
Please find below comments and observations on your manuscript. You should make changes and update the document.
1) The introduction should include ethnobotanical, entnomedicinal or ethnopharmacological references to the traditional use and medicinal effects of the plant.
### Thank you for your suggestion. So, we added more information of the traditional uses of this plant. However, this plant has not been reported for anti-diabetes and phytochemistry.
2) The authors mention that there are few studies on the plant in question. When searching, some references were found that are not in the manuscript. Therefore, reference should be made to what is currently available on the plant.
### Thank you for your recommendation. Since, the list of publications as you suggested concerning of this plant are related to agricultural sciences, they are not relevant to anti-diabetic activity. However, some articles as you mentioned can be used to fulfill the biological and traditional use details.
Lepionurus sylvestris seedling container cultivation method. By: Chen, Jinyan; Zhao, Zhiheng; Li, Kaixiang; Liang, Wenhui; Liao, Jianming; Zeng, Xiangyan. China, CN105918048 A 2016-09-07 | Language: Chinese, Database: CAplus
Seedling cultivation substrate for Lepionurus sylvestris and preparation method thereof. By: Zhao, Zhiheng; Li, Kaixiang; Liang, Wenhui; Zhu, Changsan; Liao, Jianming; Li, Baocai. China, CN108184604 A 2018-06-22 | Language: Chinese, Database: CAplus
Antimicrobial activities of some Thai traditional medical longevity formulations from plants and antibacterial compounds from Ficus foveolata. By: Meerungrueang, W.; Panichayupakaranant, P. Pharmaceutical Biology (London, United Kingdom) (2014), 52(9), 1104-1109 | Language: English, Database: CAplus and MEDLINE
### There is a report of anti-microbial activity of the L. sylvestris
Antioxidant and lifespan-extending effects of a rejuvenating thai traditional polyherbal remedy (Phy-Blica-O) in Caenorhabditis elegans. By: Chanthasri, Wipawee; Aan, Goon Jo; Singkonpong, Ngamsiri; Sudkhaw, Teerawat; Maneenoon, Katesarin; Limsuwan, Surasak; Sanpinit, Sineenart; Wetchakul, Palika; Chusri, Sasitorn. Tropical Journal of Natural Product Research (2021), 5(9), 1554-1568 | Language: English, Database: CAplus
### This publication studied some recipes which contained L. sylvestris
The first mistletoes: Origins of aerial parasitism in Santalales. By: Vidal-Russell, Romina; Nickrent, Daniel L. Molecular Phylogenetics and Evolution (2008), 47(2), 523-537 | Language: English, Database: CAplus and MEDLINE
Study on the total flavonoids content determination of the Lepionurus sylvestris BL. and its antioxidant activity. By: Zhu, Cheng-hao; Tang, Jian-min; Wei, Xiao; Gao, Li-mei; Zou, Rong; Qin, Hui-zhen; Zhu, Hong-jie, Shipin Yanjiu Yu Kaifa (2019), 40(20), 168-175 | Language: Chinese, Database: CAplus
-Can be used to fulfill total flavonoid content experiment
Lepionurus sylvestris bl. seedling raising method for promoting root system development through root cutting treatment. By: Zhu, Changsan; Chen, Jinyan; Zhao, Zhiheng; Li, Kaixiang; Liang, Wenhui; Huang, Kaishun; An, Jiacheng. China, CN110073886 A 2019-08-02 | Language: Chinese, Database: CAplus
3) All sections of the methodology should be written in detail. The methodology must be rewritten.
### Thank you, it has done by rewriting.
4) The discussion and analysis of the results can be more extensive. It should be reviewed and updated.
### Thank you, it has done by rewriting.
5) At the end of the discussion section you should include a paragraph or paragraphs with the strengths and weaknesses of your work.
### Thank you, it has done by rewriting.
6) What is the name of the new diterpene? The IUPAC name of the new compound must be included.
### - Thank you, it has done by rewriting.
Authors make changes and update the manuscript.
Regards
### Thank you for all suggestions, we tried to correct and improve our manuscript and update it already.
Best Regards,
Authors

Reviewer 2 Report
Sathianpong Phoopha et al., reported the Phytochemical investigation of Lepionurus sylvestris Blume and their effects on anti-diabetes via anti-alpha glucosidase and insulin secretagogue activities plus molecular docking.
It’s an interesting research finding and provided valid information regarding the anti-diabetic potential of Lepionurus sylvestris Blume and its phytochemical investigations.
Queries must be addressed by authors
Lepionurus sylvestris – need to provide adequate details for its pharmacological potential in introduction part
Lot of unwanted details have been provided in the abstract, so it has to be concise
Keywords section looks abbreviations part – need to provide clear information
Problem statement about the manuscript need a modification
Spell mistakes and grammatical errors have to be corrected throughout the manuscript.
I would suggest authors to remove the Table 2, since the research article focuses on anti-diabetics nature of plant and its bioactive compound.
Meanwhile, authors must perform more anti-diabetic experiments to confirm its anti-diabetic properties.
Provide the justification embryonic toxicity on zebrafish embryonic model of L. sylvestris leaf extracts. How about the experimental rat or mice model?
Author Response
Reply to Reviewer 2
Comments and Suggestions for Authors
Sathianpong Phoopha et al., reported the Phytochemical investigation of Lepionurus sylvestris Blume and their effects on anti-diabetes via anti-alpha glucosidase and insulin secretagogue activities plus molecular docking.
It’s an interesting research finding and provided valid information regarding the anti-diabetic potential of Lepionurus sylvestris Blume and its phytochemical investigations.
Queries must be addressed by authors
Lepionurus sylvestris – need to provide adequate details for its pharmacological potential in introduction part
### Thank you for your suggestion, it has done by rewriting.
Lot of unwanted details have been provided in the abstract, so it has to be concise
### Thank you for your suggestion, it has done by rewriting.
Keywords section looks abbreviations part – need to provide clear information
### Thank you for your suggestion, it has done by rewriting.
Problem statement about the manuscript need a modification
### Thank you for your suggestion, it has done by rewriting.
Spell mistakes and grammatical errors have to be corrected throughout the manuscript.
### Thank you for your suggestion, it has done by rewriting.
I would suggest authors to remove the Table 2, since the research article focuses on anti-diabetics nature of plant and its bioactive compound.
### Thank you for your suggestion. I would like to clarify that Table 2 was still kept in this manuscript although our work was focused on anti-diabetes, however this plant has not been reported of anti- cancer or cytotoxicity before. So, from the Table 2 we could see the toxicity of the plant extracts to some cancer cell lines and the safety or toxicity to the normal cell (HGF). Thus, we think this information will be useful for the other readers or researchers when they would like to consume or further study about this plant in the future.
Meanwhile, authors must perform more anti-diabetic experiments to confirm its anti-diabetic properties.
### Thank you for your suggestion, we plan to do more anti-diabetic assays in the future work and also anti-diabetic experiment in animal model and we suggest in the discussion chapter.
Provide the justification embryonic toxicity on zebrafish embryonic model of L. sylvestris leaf extracts. How about the experimental rat or mice model?
### At the beginning, we tried to do preliminary screening in basic experiment that could be guilded on anti-diabetic activity of the plant. For further study, we are planning to investigate diabetic zebrafish models that are widely used on pre-animal models. The experiment on rats or mice is the best model to clarify the systematic activity but it consumes very high cost of animal price and takes time to investigate.
### Thank you for all suggestions, we tried to correct and improve our manuscript and update it already.
Best Regards,
Authors

Reviewer 3 Report
The concept of the work is interesting and might be useful to the respective field researchers.
The abstract is too lengthy, provide highlighted information only.
Mention, which solvent was used for extraction in the abstract section
antidiabetic assay results are not mentioned in the abstract
Why author didn't perform alpha-amylase assays?
5 compounds derived from which extract (hexane, ethyl acetate, ethanol,)?
either the compound derived from all or one?
if one solvent extract is considered for further studies means, which one the author has selected and should justify on what basis?
The author may provide one more valuable keyword
the introduction section is weak, authors need to improve this section.
The author may add current clinical significance and prevalence state of diabetes in the author's country and global level in the introduction section
More valuable information about plants needs to be added to the introduction section
previous medicinal properties related reports, especially antidiabetic properties of this plant need to be added
The authors may add phytochemicals/bioactive compounds previously reported
The author should provide a novelty statement in the introduction section
provide detailed extraction methods
Other methods sections also need more detail
The discussion section needs to be more depth and add appropriate references.
Provide the full form for INS-1E. all the abbreviations should be explained in full form at the first time of entry. so the author should check the entire manuscript.
If possible author may add cytotoxicity result pictures (cell line study images)
conclusion section needs to be revised and provide valuable suggestions, and add future perspective to this study.
there are some typographical errors, authors need to check the entire manuscript.
Manuscript needs language correction.
Comments given. Manuscript needs minor language correction.
Author Response
Reply to Reviewer 3
The concept of the work is interesting and might be useful to the respective field researchers.
The abstract is too lengthy, provide highlighted information only.
### Thank you for your suggestion, it has done by rewriting.
Mention, which solvent was used for extraction in the abstract section
### Thank you for your suggestion, it has done by rewriting.
antidiabetic assay results are not mentioned in the abstract
### Thank you for your suggestion, it has done by rewriting.
Why author didn't perform alpha-amylase assays?
### Although, amylase enzyme can be digested the carbohydrate, but alpha-glucosidase in the small intestine is the key enzyme hydrolyzing oligosaccharides to monosaccharides. If we can delay or inhibit the function of alpha-glucosidase, the glucose absorption will be decreased.
5 compounds derived from which extract (hexane, ethyl acetate, ethanol,)?
### Thank you for your suggestion, it has done by rewriting to clarify the extraction and purification of isolated compounds.
either the compound derived from all or one?
### The compounds were isolated from fractionated ethanolic leaf extract only.
if one solvent extract is considered for further studies means, which one the author has selected and should justify on what basis?
### We will select ethanolic extract due to the biological activity and toxicity on zebrafish.
The author may provide one more valuable keyword
### Thank you for your suggestion, it has done by rewriting. We provided more keywords as anti-diabetes and embryonic toxicity.
the introduction section is weak, authors need to improve this section.
### Thank you for your suggestion, it has done by rewriting.
The author may add current clinical significance and prevalence state of diabetes in the author's country and global level in the introduction section
### Thank you for your suggestion, it has done by rewriting in the introduction section, however the update government information as in 2017!! Sorry.
More valuable information about plants needs to be added to the introduction section
### Thank you for your suggestion, it has done by rewriting in the introduction section.
previous medicinal properties related reports, especially antidiabetic properties of this plant need to be added
-no reports on anti-diabetic activity of this plant
The authors may add phytochemicals/bioactive compounds previously reported
-Sorry, the phytochemicals/bioactive compounds of this plant has not been reported.
The author should provide a novelty statement in the introduction section
### Thank you for your suggestion, it has done by rewriting in the introduction section.
provide detailed extraction methods
### Thank you for your suggestion, it has done by rewriting in the materials and methods section.
Other methods sections also need more detail
### Thank you for your suggestion, it has done by rewriting in the materials and methods section.
The discussion section needs to be more depth and add appropriate references.
### Thank you for your suggestion, it has done by rewriting in the discussion section.
Provide the full form for INS-1E. all the abbreviations should be explained in full form at the first time of entry. so the author should check the entire manuscript.
### Thank you for your suggestion, it has done by rewriting.
If possible author may add cytotoxicity result pictures (cell line study images)
### Sorry for that we did not have the result pictures, we analyzed only the data . Next study we will do it, thank you for your suggestion.
conclusion section needs to be revised and provide valuable suggestions, and add future perspective to this study.
### Thank you for your suggestion, it has done by rewriting in conclusion section.
there are some typographical errors, authors need to check the entire manuscript.
### Thank you for your suggestion, it has done by checking and rewriting.
Manuscript needs language correction.
### Thank you for your suggestion, it has done by checking and rewriting.
Comments on the Quality of English Language
### Thank you for your suggestion, it has done by checking and rewriting.
Comments given. Manuscript needs minor language correction.
### Thank you for your suggestion, it has done by checking and rewriting.
### Thank you for all suggestions, we tried to correct and improve our manuscript and update it already.
Best Regards,
Authors

Round 2
Reviewer 1 Report
Dear authors thank you for submitting your updated manuscript.
You should take into account the following observations and comments in order to update the manuscript.
1) Include the drug:solvent ratio used in each extraction process and include the percentage yields, e.g. 400 g : 4000 mL Hexane, i.e. (1:10) and so on for other extractions carried out with other solvents. In relation to the yields with the extracts to include them as a percentage, e.g. 1.2% for hexane, 7.8% for ethyl acetate, etc., either in text or table form.
2) The amount in ug, mg or grams of the compounds obtained that were isolated, purified, elucidated and identified should be written/indicated.
3) For the new compound, which was named according to IUPAC rules, you can also give a common name to your choice/criteria.
Authors make the changes and congratulations on your work.
Best regards
Reviewer
Author Response
Reply to Reviewer 1-Round 2
Comments and Suggestions for Authors
Dear authors thank you for submitting your updated manuscript.
You should take into account the following observations and comments in order to update the manuscript.
1) Include the drug:solvent ratio used in each extraction process and include the percentage yields, e.g. 400 g : 4000 mL Hexane, i.e. (1:10) and so on for other extractions carried out with other solvents. In relation to the yields with the extracts to include them as a percentage, e.g. 1.2% for hexane, 7.8% for ethyl acetate, etc., either in text or table form.
### Thank you for your suggestion. In our extraction method as used in this study was maceration. For maceration extraction, you can soak the required amount of medicinal plant raw material in a maceration tank with a solvent at room temperature for seven days with periodic stirring. We could repeat it several time in order to increase the percentage yield, so the crude extract yield depends on the time of extraction. The volume of the solvent needs to cover all samples, it means the solvent level must be higher than the sample in the tank. So, we are sorry that we recorded only the extraction time, that necessary for the yield.
### We added the percentage yield of all crude extracts as you suggested already.
2) The amount in ug, mg or grams of the compounds obtained that were isolated, purified, elucidated and identified should be written/indicated.
### It has done.
3) For the new compound, which was named according to IUPAC rules, you can also give a common name to your choice/criteria.
### Thank you for your suggestion. So, we have done as the new diterpene derivative was named according to IUPAC rules as 15-16-epoxy-neo-cleoda-3,7(20),13(16),14-tetraene-12,17:18,19-diolide or was named as lepionurodiolide
Authors make the changes and congratulations on your work.
### Thank you for all suggestions, we tried to correct and improve our manuscript and update it already.
Best Regards,
Authors

Reviewer 2 Report
The revised version of manuscript is acceptable for publication.
Author Response
Reply to Reviewer 2 - Round 2
Comments and Suggestions for Authors
The revised version of manuscript is acceptable for publication.
### Thank you very much for your suggestions.
Best Regards,
Authors

Reviewer 3 Report
The authors addressed the comments satisfactorily
Minor language corrections need
Author Response
Reply to Reviewer 3 - Round 2
Comments and Suggestions for Authors
The authors addressed the comments satisfactorily
### Thank you very much for your suggestions.
Comments on the Quality of English Language
Minor language corrections need
### Thank you for your suggestion. So, we tried our best to correct English language throughout our manuscript.
Best Regards,
Authors
